# DEAGLE: TOKEN TREE WITH DYNAMIC DEPTH WILL FURTHER BENEFIT THE SPECULATIVE DECODING

## ABSTRACT

Large Language Models (LLMs) have shown remarkable capabilities in text generation, but they also suffer from high token-by-token latency due to the nature of autoregressive decoding. Speculative decoding (SD) mitigates this by using the draft-then-verify framework, making it possible to generate multiple tokens in a single LLM forward pass. However, existing state-of-the-art SD frameworks typically generate token trees with a fixed depth, which brings unnecessary computation and suboptimal speedup across diverse datasets. In this work, we introduce DEAGLE, a lightweight and training-free extension to EAGLE-3 that enables adaptive-depth speculative decoding through context-aware token-tree monitoring. We provide the first formal proof that draft model confidence serves as an unbiased estimator of token-level acceptance, generalizing empirical observations from prior EAGLE-2 work to EAGLE-3. Furthermore, we show that the product of draft confidences along a token path, the survival probability, can be a good heuristic for full-branch acceptance. Based on this insight, DEAGLE introduces a voting-based early stopping mechanism that monitors the survival probability sum of the top-k leaves, survival momentum, and the expected accept length for the whole token tree (estimated via survival probability expectation). These factors are jointly used to determine when to stop tree expansion. DEAGLE can be integrated into EAGLE-3 without retraining or architectural changes. Experiments on Vicuna 13b, Llama3-8b, and Llama3-70b demonstrate that DEAGLE achieves further speedup over EAGLE-3 and enables more robust acceleration across different datasets and token tree depths.

## 1 INTRODUCTION

Large Language Models (LLMs) have demonstrated remarkable performance in a variety of natural language processing (NLP) tasks OpenAI (2023). However, their practical use is limited by inference latency. This latency originates from the nature of Auto-Regressive decoding, which requires generating $n$ tokens through n sequential forward passes. The resulting sequential computation reduces the effectiveness of parallel hardware and leads to memory-bandwidth constraints that drive up computational costs Leviathan et al. (2023).

To address this challenge, a range of acceleration strategies have been developed to improve the efficiency of computational resources. Given that the inference latency of large models is constrained by memory bandwidth rather than arithmetic computation Leviathan et al. (2023), speculative decoding (SD) has emerged as a compelling approach. SD utilizes a smaller model (the "draft model") to propose multiple tokens (draft tokens) with small overhead, then the original larger "target model" verifies those proposed draft tokens in parallel in batches with a single forward pass. By this parallel token verification process, speculative decoding significantly increases the inference speed while preserving the exact distribution and quality of model outputs.

Recent developments in speculative decoding frameworks have improved efficiency by shifting from single-sequence drafts to multi-branch speculative trees, as seen in architectures such as Medusa Cai et al. (2024) and SpecInfer Miao et al. (2024). These models introduced the "tree attention" mechanisms that apply topology-aware causal masks to enforce strict parent-child attention within each candidate tree, which prevents interference across branches. As a result, the verification process extends from a single sequence to multiple speculative branches. This method increases the Mean

Accepted Token (MAT), which measures the average number of draft tokens successfully verified per decoding step, and reduces GPU memory bandwidth consumption by increasing the computational density of each forward pass.

Among recent methods, the Eagle models (Eagle-1 Li et al. (2024a), Eagle-2 Li et al. (2024b), Eagle-3 Li et al. (2025)) greatly improved speculative decoding on the draft tokens' acceptance rates, and thus became the current state-of-the-art SD model. Eagle 1 introduced the notion of feature-level auto regression and created the basic structure used in later Eagle models. Eagle-2 extended this approach with an empirical finding that the token tree confidence scores from the draft model show a positive relation with token acceptance rates from the LLM. With that finding, Eagle-2 introduced a Dynamic Token Tree Expansion mechanism, making the token tree branches vary based on the context. Eagle-3 further refined the framework by removing the feature loss component and retaining only the classification loss. This modification was specifically designed to eliminate the constraint brought by the feature loss and make the scaling law achieve, resulting in a tighter bound compared to Eagle-2 on the correlation between token tree confidence scores and acceptance rates (shown in Section 3).

Despite these improvements, one major limitation remains. Eagle-2 sets a fixed depth token tree for speculative decoding, which inevitably introduces suboptimal tuning and compromises efficiency across various application scenarios. A more optimal token tree depth needs to be adaptive and context-aware. The fixed depth approach cannot balance the gain from longer accepted token sequences and therefore wastes extra draft model forward passes. The fixed-depth token tree might severely under-utilize the speculative potential of the draft model for simple sentences, whereas drafting for complex sentences might suffer unnecessary computations from short acceptance length.

In this paper, we introduce **DEAGLE** (Dynamic EAGLE), a speculative decoding framework based on EAGLE-3 using the depth-adaptive token tree. Unlike earlier methods that use a fixed depth hyperparameter, DEAGLE dynamically adjusts the tree depth for each inference run based on a voting mechanism. We first give a formal proof for the positive correlation between confidence scores and acceptance rates initially observed in Eagle-2 in Section 3.1. In Section 3.2, we extend this proof to Eagle-3 (as formalized in Equation 19), confirming that the architectural modification (i.e., the removal of feature lo ss) results in a smaller bound and a more robust correlation, thereby validating the theoretical consistency across the Eagle lineage. This forms the basis for our training-free prediction of the optimal dynamic depth for Eagle-3 with minimal additional overhead.

Specifically, **DEAGLE** makes the following major contributions:

- **Confidence-acceptance equivalence**: We formally prove that the draft confidence scores $c_t$ and the token acceptance probabilities $\alpha_t$ are bounded in Eagle-2 and Eagle-3 structures (Section 3.2).

- **Token Tree Expectation as heuristic**: We introduce the notion of using survival probability expectation as an estimation for the average acceptance length of the given token tree, which becomes a good heuristic for the draft model to stop building the token tree (Section 3.3).

- **Voting-based depth control**: **DEAGLE** predicts the tree depth based on evaluation of three factors: (i) top-$k$ survival probability sum $S_d$, (ii) survival momentum $\rho_d$, and (iii) expected acceptance length $E_d$ triggers early termination when expansion efficiency drops (Section 3.4). We performed extensive experiments and compared the speedup ratio of our DEAGLE and EAGLE-3 across different models, temperatures, and datasets to verify the effectiveness of this approach in Section 4.

## 2 RELATED WORKS

**LLM Inference Acceleration and Speculative Decoding Foundations.** Large-language-model inference acceleration has been approached through quantization Frantar et al. (2023); Dettmers et al. (2022), pruning Ma et al. (2023), and knowledge distillation Hinton et al. (2014). As these methods often trade model quality for speed, speculative decoding emerged as a lossless acceleration technique: Stern et al. introduced blockwise parallel decoding Stern et al. (2018); Leviathan et al. and Chen et al. later formalized the *draft-then-verify* paradigm with rigorous distribution-preservation guarantees Leviathan et al. (2023); Chen et al. (2023). Most of these meth-

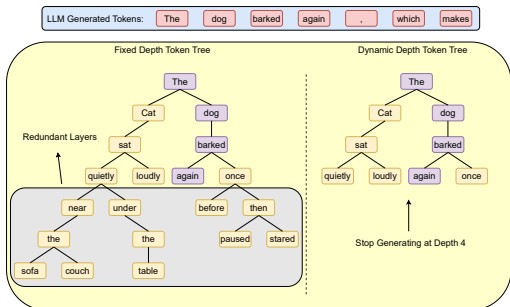

Figure 1: Fixed Depth Token Tree vs. Dynamic Depth Token Tree. The red tokens are the ground truth tokens from LLM for verification, and the purple tokens are the correct token sequence generated by the draft model. In the token tree with a dynamic depth control mechanism, there will be three fewer draft model forward passes than the regular fixed depth token tree.

ods perform a strict greedy sampling, choosing only the most probable draft token, and greatly limit the potential of the draft model. A more flexible tree-based approach was then proposed.

**Tree-Based Speculation and Feature-Level Innovations.** SpecInfer pioneered token-tree verification and achieved 57–97% higher verification success Miao et al. (2024). Medusa adopted multi-head prediction to generate future tokens in parallel Cai et al. (2024), while Hydra introduced sequentially-dependent draft heads Ankner et al. (2024). The Eagle architectures made feature-level breakthroughs: Eagle-1 employed feature autoregression for higher quality draft token generation Li et al. (2024a), Eagle-2 built dynamic draft trees with draft model confidence score to select more valuable draft sequence for verification Li et al. (2024b), and Eagle-3 removed the feature loss to eliminate the scaling law constraint and integrate features from more decoder layers to further improve the draft token quality Li et al. (2025).

**Adaptive Control and Depth Limitations.** Other than EAGLE-2, several recent works have attempted to improve speculative decoding by adaptively controlling draft depth or candidate length. Brown et al. (2024) adjusts draft depth at each step according to confidence heuristics, but it relies on simple rule-based thresholds without theoretical guarantees. Lu et al. (2024) uses MLPs to decide whether to build the next depth of a fork-shaped token tree, which incurs too much overhead and can't achieve similar acceleration as EAGLE. Wang et al. (2025) design OPT-Tree that prunes branches dynamically with fixed depths. Mamou et al. (2024); Huang et al. (2025); Liu et al. (2025) can only be used to predict the single sequence length and cannot be used to predict the depth of the token tree. However, all of those methods are either failed to beat the speedup brought by or were not compatible with EAGLE-2, which has a fixed-depth token tree. Usually, a fixed-depth token tree might keep generating tokens until the maximum depth, even when draft tokens' quality is low and unlikely to be accepted (Fig 1). This produces redundant branches that do not help the final output. These extra layers waste computation and requires more memory operations, which slows decoding and reduces the overall efficiency. Although Eagle-3 is the current state-of-the-art in speculative decoding fields, it still wastes computation by sticking to a fixed depth token tree that blindly expands unnecessary branches.

## 3 APPROACH

### 3.1 FEATURE-LEVEL ALIGNMENT AND BOUNDED KL DIVERGENCE IN EAGLE

In speculative decoding frameworks, the draft model must generate token proposals whose distributions closely match those of the LLM. That draft-LLM feature alignment become even more important for models like EAGLE-1 and EAGLE-2, which generate and reuse hidden features in an autoregressive way to improve the draft feature quality. EAGLE models ensure tight alignment via a feature regression loss $f_{reg}$ and a Token Classification Loss $f_{cls}$):

$$L_{\text{reg}} = \text{Smooth-}L1\big(f_{i+1}, \hat{f}_{i+1}\big)$$
$$L_{\text{cls}} = \text{CrossEntropy}\big(p_{i+1}, \hat{p}_{i+1}\big) \tag{1}$$
$$L = L_{\text{reg}} + w_{\text{cls}} L_{\text{cls}}$$

Here, $f_{i+1}$ is the hidden feature from the LLM for token $i + 1$, and $\hat{f}_{i+1}$ is the predicted feature via autoregressive draft model with input of the previous draft hidden feature $\hat{f}_{i+1}$ and draft token embedding $\hat{e}_i$. The classification loss further aligns the LLM token logits distribution $p_{i+1}$ and draft token logits $\hat{p}_{i+1}$ after the shared LLM head with weight $W_{\text{out}}$ by minimizing their cross-entropy. The final L integrated both losses with the term $w_{cls}$ to balance the effect of $L_{cls}$

**Feature Alignment Implies Logit Closeness.**   Based on equation (1), during the training, Eagle is trying to minimize the smooth L1 loss between the draft feature and the LLM feature. Therefore, the draft feature $\hat{f}_t$ is trained to approximate the target feature of LLM $f_t$. Formally,

$$\|f_t - \hat{f}_t\| \leq \epsilon_f \tag{2}$$

holds for each autoregressive step $t$, where $\epsilon_f$ represents the feature-level error bewteen the draft feature and the LLM feature. Since $W_{\text{out}}$ is reused for both models, the corresponding draft and LLM logits satisfy:

$$\|\ell_t^{(llm)} - \ell_t^{(d)}\| = \|W_{\text{out}}(\hat{f}_t - f_t)\| \leq \|W_{\text{out}}\| \cdot \epsilon_f \tag{3}$$

Given the Lipschitz continuity of softmax and standard KL upper bounds in terms of logit differences, we have:

$$\text{KL}\big(p_t^{(llm)} \,\|\, p_t^{(d)}\big) \leq \frac{1}{2}\|W_{\text{out}}\|^2 \epsilon_f^2 \tag{4}$$

In addition, the classification loss:

$$L_{\text{cls}} = \text{CE}(p_t^{(llm)}, p_t^{(d)}) = H(p_t^{(llm)}) + \text{KL}(p_t^{(llm)} \,\|\, p_t^{(d)}) \tag{5}$$

provides explicit supervision over the token-level output distributions. Since $L_{\text{cls}}$ minimizes the KL divergence, its effect indirectly complements the KL term from (4). Then, the actual divergence may be further tightened:

$$\text{KL}(p_t^{(llm)} \,\|\, p_t^{(d)}) \leq max(0, \tfrac{1}{2}\|W_{\text{out}}\|^2 \epsilon_f^2 - \delta_{\text{cls}}), \tag{6}$$

where $\delta_{\text{cls}} \propto w_{\text{cls}} L_{\text{cls}}$.

As a result, at each draft step $t$, the divergence between the draft and LLM token distributions is deterministically bounded.

**Autoregressive Prefix Drift and Cumulative Bound.**   EAGLE expands tokens autoregressively: each draft token affects the next prefix and hence the next feature prediction. However, because $f_t \approx \hat{f}_t$ at each step, the prefix mismatch grows slowly. By invoking feature regression at every step, we maintain:

$$\|\delta_t\| := \|p_{<t}^{(llm)} - p_{<t}^{(d)}\| \text{ grows at most linearly with } t, \tag{7}$$

where $\delta_t$ represents the logits-level difference at time t. As a result, the cumulative logit- and distribution-level divergence remains bounded, scaling sub-quadratically with depth. Empirically, the difference between draft and LLM predictions remains small even for trees of depth up to 8.

Through repeated application of feature-level regression and a shared LM head, EAGLE's draft model maintains a bounded gap to the LLM logits at every step, and therefore, the KL divergence between draft and LLM token distributions is provably bounded.

3.2   CONFIDENCE SCORE AS A HEURISTIC FOR ACCEPTANCE PROBABILITY

Building upon the bounded divergence established in Section 3.1, we now formalize why the draft confidence score can reliably estimate the probability that a draft token is accepted by the LLM.

**Definitions.**   Let the draft model output token $\hat{y}$ and confidence score $c$ at step $t$ be

$$\hat{y}_t = \arg\max_y p_t^{(d)}(y), \quad c_t = p_t^{(d)}(\hat{y}_t) \tag{8}$$

and define the acceptance indicator as

$$a_t := \mathbb{I}\left[\hat{y}_t = \arg\max_y p_t^{(llm)}(y)\right], \quad a_t \in \{0, 1\}, \tag{9}$$

where y is a variable for choosing logits from the logits distribution p for both the draft model and LLM. Then, the acceptance probability is given by the expectation over the indicator random variable $a_t$:

$$\alpha_t := \mathbb{E}[a_t] \tag{10}$$

**Bounding the Acceptance Gap.**    From Section 3.1, the KL divergence between the draft and LLM distributions is bounded by:

$$D_{\text{KL}}(p_t^{(llm)} \parallel p_t^{(d)}) \leq max(0, \tfrac{1}{2}M^2\epsilon_f^2 - \delta_{\text{cls}}), \tag{11}$$

where $M = \|W_{\text{out}}\|$ is the norm of the shared output projection, and $\epsilon_f$ is the feature alignment error. By applying Pinsker's inequality Cover & Thomas (2006), this leads to a total variation $TV$ bound:

$$\|p_t^{(llm)} - p_t^{(d)}\|_{\text{TV}} \leq \sqrt{\tfrac{1}{2}D_{\text{KL}}} \leq \sqrt{max(0, \tfrac{1}{2}M^2\epsilon_f^2 - \delta_{\text{cls}})} \tag{12}$$

**Relating Confidence to Acceptance.**    By coupling arguments, for any token $y$:

$$|p_t^{(llm)}(y) - p_t^{(d)}(y)| \leq \|p_t^{(d)} - p_t^{(llm)}\|_{\text{TV}} \tag{13}$$

—so in particular, if $y$ is the most probable token $\hat{y}_t$,

$$|p_t^{(llm)}(\hat{y}_t) - c_t| \leq \sqrt{max(0, \tfrac{1}{2}M^2\epsilon_f^2 - \delta_{\text{cls}})} \tag{14}$$

Although $\hat{y}_t$ is not guaranteed to be the top-1 token under $p_t^{(llm)}$, the fact that $p_t^{(d)}$ is close to $p_t^{(llm)}$ implies that $\hat{y}$ is still likely to receive relatively high probability under the LLM. In particular, when both distributions are sharp—as is typical under low-temperature decoding—$p_t^{(llm)}(\hat{y}_t)$ serves as a soft estimator for the acceptance probability $\alpha_t$.

$$|\alpha_t - c_t| \leq \sqrt{max(0, \tfrac{1}{2}M^2\epsilon_f^2 - \delta_{\text{cls}})}, \quad \alpha_t \approx p_t^{(llm)}(\hat{y}_t) \tag{15}$$

Hence, the draft confidence score $c_t$ will give a close approximation when the draft model and LLM distributions are well-aligned.

**Consistency with Prior Work.**    EAGLE-2 empirically demonstrated that the draft confidence score $c_t$ aligns well with the true acceptance probability $\alpha_t$. Now, we provide a theoretical justification for this phenomenon based on its feature-level alignment mechanism and the resulting bounded divergence between the draft and LLM distributions.

Under the bounded KL divergence regime, the LLM-assigned probability $p_t^{(llm)}(\hat{y}_t)$ for the draft-selected token serves as a soft estimator for the acceptance probability $\alpha_t = \mathbb{E}[a_t]$. And now, we have the guaranteed lower bound under the low-temperature scenario:

$$\boxed{|\alpha_t - c_t| \leq \sqrt{max(0, \tfrac{1}{2}M^2\epsilon_f^2 - \delta_{\text{cls}})}} \tag{16}$$

Since EAGLE-3 removes the feature loss, we measure the discrepancy at only the logits distribution level. Let

$$\epsilon_t := \left\|p_t^{(llm)} - p_t^{(d)}\right\|_{\text{TV}} \tag{17}$$

By Pinsker's inequality and (5),

$$\epsilon_t \leq \sqrt{\tfrac{1}{2}D_{\text{KL}}} = \sqrt{\tfrac{1}{2}(L_{cls} - H(p_t^{(llm)}))} \tag{18}$$

Therefore, the gap between the draft confidence and the acceptance probability is bounded without involving any projection norm:

$$\boxed{|c_t - \alpha_t| \leq \epsilon_t \leq \sqrt{\tfrac{1}{2}(L_{cls} - H(p_t^{(llm)}))}} \tag{19}$$

which shows that $c_t$ becomes an even more accurate soft estimator for $\alpha_t$ as the loss no longer amplified by the projection and the quadratic term from softmax.

## 3.3 Survival Probability and Expected Acceptance Length

Having shown in Section 3.2 that draft confidence $c_t$ closely estimates the acceptance probability $\alpha_t$, we now extend this to the token sequence. Specifically, we justify the use of cumulative survival probability as a heuristic for the probability that a whole branch is accepted as EAGLE-2 did, which makes it possible to evaluate speculative decoding trees and estimate the expected number of accepted tokens.

**Survival Probability over a Token Branch.** Let a token branch of depth $L$ be represented as a sequence $\hat{y}_{1:L}$, where $\hat{y}_t = \arg\max_y p_t^{(d)}(y)$. Define the survival probability at depth $t$ as:

$$s_t := \prod_{i=1}^{t} c_i \tag{20}$$

Since each $c_i \approx \alpha_i$, this product approximates the joint probability that all draft tokens along the branch until depth $t$ are accepted by the LLM. That is:

$$\Pr[\text{branch } \hat{y}_{1:t} \text{ is accepted}] \approx s_t \tag{21}$$

**Tail-Sum Trick for Expected Acceptance Length.** Let $A$ denote the random variable for the number of consecutively accepted tokens in the branch. Then the expectation of $A$ can be computed via:

$$\mathbb{E}[A] = \sum_{t=1}^{L} \Pr[A \geq t] \approx \sum_{t=1}^{L} s_t \tag{22}$$

This is a standard identity known as the "tail-sum formula" Ross (2018). Since each $s_t$ captures the marginal likelihood that the branch survives until step $t$, the total sum gives the expected acceptance length over the full branch.

Then, the expectation with tail-sum of $s_t$ can serve as a reliable heuristic for the expected number of tokens accepted by the LLM.

$$\boxed{\mathbb{E}[A] \approx \sum_{t=1}^{L} \prod_{i=1}^{t} c_i} \tag{23}$$

## 3.4 Tree Depth Control via Voting on Survival Signals

Having defined the survival scores $s_t$ and the expected acceptance length in Sections 3.2–3.3, we now describe how DEAGLE uses three complementary factors to dynamically control token tree expansion:

**Probability Sum of Top-$k$ Leaves.** At each depth $d$, consider the set $\mathcal{L}_d$ of current leaf nodes. We compute:

$$S_d := \sum_{i=1}^{k} s_d^{(i)} \tag{24}$$

where $s_d^{(1)} \geq \cdots \geq s_d^{(k)}$ are the top-$k$ survival probabilities. A low $S_d$ indicates that most branches are unlikely to survive, which means that draft tokens from further expansions are unlikely to be accepted.

**Survival Momentum ($\rho_d$).** We track the relative drop in top-k survival probability between depths:

$$\rho_d := \frac{S_{d-1}}{S_d} \tag{25}$$

More than one sharp decline ($\rho_d \ll 0.6$) indicates that branches are rapidly "dying off," so deepening further is unlikely to yield accepted paths.

**Expected Accept Length Bound.** From Section 3.3, the expected number of accepted tokens in a token tree can be estimated by aggregating the joint survival probabilities of all active leaf nodes at depth $d$:

$$E_d := \sum_{\ell \in \text{Leaf}(d)} s_\ell \tag{26}$$

Let $D_{\max} := \lceil E_d \rceil$ denote the expected acceptance length horizon. Once the tree reaches depth $d \geq D_{\max}$, we assume the draft sequence has likely reached the maximal number of LLM-approved tokens, and further depth expansions are unnecessary.

**Combined Voting System ("DEAGLE").** To determine whether to continue expanding the token tree at depth $d$, we introduce a three-component voting mechanism. The expansion stops if at least two out of the following three conditions are satisfied:

- **Low Probability:** $S_d < \tau_S$, where $S_d$ denotes the total survival probability of top-$k$ leaf nodes at depth $d$.

- **Sharp Momentum Decay:** $\rho_d < \tau_\rho$ for two times, where $\rho_d := \frac{S_d}{S_{d-1}}$ reflects the rate of survival mass decay compared to the previous depth.

- **Acceptance Length Saturation:** $d \geq \lceil E_d \rceil$, where $E_d$ is the expected number of tokens accepted by the LLM, as estimated in Section 3.3.

or the token tree reached the maximum depth.

## 4 EXPERIMENTS

| Depth | MT-Bench EAGLE-3 | DEAGLE | Human Eval EAGLE-3 | DEAGLE | Gsm8k EAGLE-3 | DEAGLE | Alpaca EAGLE-3 | DEAGLE |
|---|---|---|---|---|---|---|---|---|
| | | | | Temperature=0 | | | | |
| 6 | 4.12x | **4.20x** | 4.63x | **4.67x** | 4.39x | **4.44x** | 4.18x | **4.25x** |
| 8 | 4.37x | **4.41x** | 5.13x | 5.13x | 4.55x | **4.66x** | 4.37x | **4.45x** |
| 10 | 4.37x | **4.52x** | 5.37x | **5.41x** | 4.47x | **4.58x** | 4.32x | **4.48x** |
| 12 | 4.30x | **4.56x** | 5.42x | **5.60x** | 4.27x | **4.46x** | 4.14x | **4.46x** |
| 14 | 4.16x | **4.54x** | 5.33x | **5.61x** | 4.03x | **4.30x** | 4.00x | **4.44x** |
| 16 | 3.99x | **4.53x** | 5.15x | **5.58x** | 3.82x | **4.23x** | 3.74x | **4.34x** |
| 18 | 3.78x | **4.45x** | 5.01x | **5.61x** | 3.60x | **4.20x** | 3.52x | **4.38x** |
| | | | | Temperature=1 | | | | |
| 6 | **3.62x** | 3.59x | 4.11x | **4.20x** | **3.71x** | 3.69x | 3.65x | **3.74x** |
| 8 | 3.69x | 3.69x | **4.36x** | 4.34x | 3.67x | **3.78x** | 3.57x | **3.75x** |
| 10 | 3.56x | **3.77x** | **4.52x** | 4.33x | 3.57x | **3.63x** | 3.57x | **3.66x** |
| 12 | 3.39x | **3.65x** | 4.16x | **4.42x** | 3.19x | **3.53x** | 3.31x | **3.61x** |
| 14 | 3.35x | **3.66x** | 4.17x | **4.41x** | 3.08x | **3.50x** | 3.07x | **3.47x** |
| 16 | 3.12x | **3.71x** | 3.60x | **4.39x** | 2.97x | **3.39x** | 2.82x | **3.54x** |
| 18 | 3.07x | **3.58x** | 3.72x | **4.50x** | 2.77x | **3.47x** | 2.76x | **3.49x** |

Table 1: Speedup comparison between EAGLE-3 and DEAGLE on **Vicuna-13B** across varying tree depths and decoding temperatures. DEAGLE consistently matches or exceeds the performance of EAGLE-3 under both greedy decoding ($T = 0$) and high-entropy sampling ($T = 1$) on all four benchmarks. Results are averaged over 80 prompts per task.

| Depth | MT-Bench EAGLE-3 | DEAGLE | Human Eval EAGLE-3 | DEAGLE | Gsm8k EAGLE-3 | DEAGLE | Alpaca EAGLE-3 | DEAGLE |
|---|---|---|---|---|---|---|---|---|
| | | | | Temperature=0 | | | | |
| 6 | 3.73x | **3.75x** | **4.02x** | 3.91x | **3.91x** | 3.89x | **3.87x** | 3.86x |
| 8 | 3.85x | **3.95x** | 4.28x | **4.30x** | 4.01x | 4.01x | 4.11x | **4.13x** |
| 10 | 3.80x | **3.96x** | 4.31x | **4.52x** | 3.84x | **4.06x** | 4.01x | **4.13x** |
| 12 | 3.53x | **3.92x** | 4.19x | **4.41x** | 3.62x | **3.90x** | 3.85x | **4.17x** |
| 14 | 3.40x | **3.90x** | 3.89x | **4.37x** | 3.39x | **3.86x** | 3.55x | **4.14x** |
| 16 | 3.16x | **3.89x** | 3.71x | **4.36x** | 3.18x | **3.91x** | 3.30x | **4.08x** |
| 18 | 3.01x | **3.94x** | 3.57x | **4.38x** | 3.01x | **3.88x** | 3.20x | **4.08x** |
| | | | | Temperature=1 | | | | |
| 6 | **2.68x** | 2.67x | 3.36x | **3.44x** | 3.29x | 3.24x | 3.24x | **3.27x** |
| 8 | 2.62x | **2.77x** | 3.41x | **3.50x** | **3.27x** | 3.22x | 3.19x | **3.26x** |
| 10 | 2.48x | **2.65x** | 3.45x | **3.55x** | 3.01x | **3.27x** | 3.07x | **3.21x** |
| 12 | 2.31x | **2.79x** | 3.29x | **3.52x** | 2.88x | **3.05x** | 2.86x | **3.04x** |
| 14 | 2.31x | **2.76x** | 3.22x | **3.55x** | 2.60x | **3.09x** | 2.71x | **3.08x** |
| 16 | 2.07x | **2.65x** | 2.89x | **3.52x** | 2.44x | **3.13x** | 2.58x | **3.05x** |
| 18 | 1.98x | **2.67x** | 2.82x | **3.52x** | 2.35x | **3.15x** | 2.34x | **3.10x** |

Table 2: Speedup comparison between EAGLE-3 and DEAGLE on **Llama3-8B**

## 4.1 4.1 EXPERIMENT SETUP

Following the setup of EAGLE-3, We perform experiments with DEAGLE on **Vicuna-13B** Chiang et al. (2023), **LLaMA3-8B**, and **LLaMA3-70B** Grattafiori et al. (2024) with their pre-trained eagle weights. We conducted experiments on four widely used benchmarks to assess decoding efficiency and robustness across various generation tasks. **MT-Bench** Zheng et al. (2023) is a multi-turn dialogue benchmark designed to evaluate alignment and conversation ability. **HumanEval** Chen et al. (2021) is a Python code generation benchmark. **GSM8K** Cobbe et al. (2021) is a grade-school math reasoning benchmark requiring step-by-step numeric generation. **Alpaca** Taori et al. (2023) consists of general instruction-following prompts derived from self-instruct methods and covers a broad range of open-ended tasks. For each benchmark, similar to the EAGLE setup, we randomly choose 80 prompts during evaluation and experimented with them for both EAGLE and DEAGLE under identical hyperparameter settings to ensure a fair comparison. Experiments for Vicuna-13B and Llama3-8B were conducted on a single NVIDIA A100 GPU, and the experiment on Llama3-70B was conducted on two A100 GPUs. We calculated the average wall time of 80 prompts as our final results. DEAGLE uses the same token tree structure as EAGLE-3, with adaptive tree depth control based on the voting strategy described in Section 3.4. The threshold of survival probability sum and survival momentum decay ratio is set to $\tau_S = 0.15$ and $\tau_\rho = 0.6$ for all models, decoding temperatures, and benchmarks.

| Depth | MT-Bench | | Human Eval | | Gsm8k | | Alpaca | |
|---|---|---|---|---|---|---|---|---|
| | EAGLE-3 | DEAGLE | EAGLE-3 | DEAGLE | EAGLE-3 | DEAGLE | EAGLE-3 | DEAGLE |
| Temperature=0 | | | | | | | | |
| 6 | 4.07x | **4.08x** | 4.56x | **4.58x** | **4.42x** | 4.41x | 4.25x | **4.27x** |
| 8 | 4.32x | **4.33x** | 5.20x | **5.23x** | **4.79x** | 4.78x | 4.68x | **4.72x** |
| 10 | 4.37x | **4.45x** | 5.43x | **5.48x** | 4.80x | **4.85x** | 4.78x | **4.85x** |
| 12 | 4.28x | **4.42x** | 5.37x | **5.49x** | 4.70x | **4.82x** | 4.76x | **4.87x** |
| 14 | 4.20x | **4.44x** | 5.26x | **5.45x** | 4.59x | **4.78x** | 4.64x | **4.83x** |
| 16 | 4.09x | **4.40x** | 5.12x | **5.44x** | 4.47x | **4.77x** | 4.53x | **4.82x** |
| 18 | 3.99x | **4.41x** | 4.99x | **5.43x** | 4.36x | **4.78x** | 4.38x | **4.83x** |
| Temperature=1 | | | | | | | | |
| 6 | 3.78x | **3.80x** | 4.17x | **4.25x** | 4.14x | **4.15x** | 4.17x | 4.17x |
| 8 | 3.99x | **4.05x** | 4.71x | **4.71x** | 4.42x | **4.45x** | 4.49x | **4.62x** |
| 10 | 3.99x | **4.09x** | 4.82x | **4.91x** | 4.44x | **4.47x** | 4.58x | **4.67x** |
| 12 | 4.01x | **4.11x** | 4.82x | **4.87x** | 4.41x | **4.49x** | 4.53x | **4.61x** |
| 14 | 3.82x | **4.06x** | 4.71x | **4.90x** | 4.23x | **4.40x** | 4.43x | **4.56x** |
| 16 | 3.78x | **4.04x** | 4.50x | **4.77x** | 4.15x | **4.41x** | 4.24x | **4.57x** |
| 18 | 3.71x | **3.99x** | 4.43x | **4.85x** | 3.99x | **4.38x** | 4.13x | **4.58x** |

Table 3: Speedup comparison between EAGLE-3 and DEAGLE on **Llama3-70B**

## 4.2 4.2 EFFECTIVENESS OF DEPTH CONTROL MECHANISM

We compare the speedup performance of DEAGLE and EAGLE-3 across different tree depths, decoding temperatures, and base models. Results are summarized in Tables 1, 2, and 3, where the acceleration of the original baseline LLM is set to 1× to compare with. DEAGLE consistently achieves equal or higher speedup than EAGLE-3 across almost all settings and successfully maintains the speedup ratio as we increase the maximum depth. On **Vicuna-13B**, DEAGLE outperforms EAGLE-3 on all four benchmarks. Under greedy decoding setting($T = 0$), DEAGLE reaches up to **5.61×** speedup on HumanEval at depth 14, compared to EAGLE-3's 5.33×. For GSM8K and MT-Bench, DEAGLE's speedup ratio remains ahead across all depths and is not affected by the increasing tree depth. Under high-temperature decoding ($T = 1$), both architectures yield lower speedups due to flatter logits distribution. In that scenario, EAGLE-3 shows a higher decreasing rate of speedup with the increase of tree depth. The dynamic depth control scheme here shows its value as the DEAGLE speedup ratio remains stable with varying depths. On **LLaMA3-8B** and **LLaMA3-70B**, DEAGLE shows its superiority in the greedy sampling scenario. The speedup ratio becomes stable after depth 12 with a small amount of change for all benchmarks. A similar trend is also shown under high temperature setting, where DEAGLE's adaptive stopping mechanism avoids over-expanding low-quality branches to make the overall draft-then-verify process more efficient. Unlike EAGLE-3, the depth control trick in DEAGLE makes the tree depth no longer a sensitive hyperparameter that remains unknown when running the model on a new dataset.

## 4.3 4.3 ANALYSIS ON REDUCED DRAFT MODEL FORWARD PASSES

We quantify the benefit brought by the dynamic depth control algorithm by comparing the extra forward passes of the draft model for both EAGLE-3 and DEAGLE at $T = 0$ with Llama3-8B.

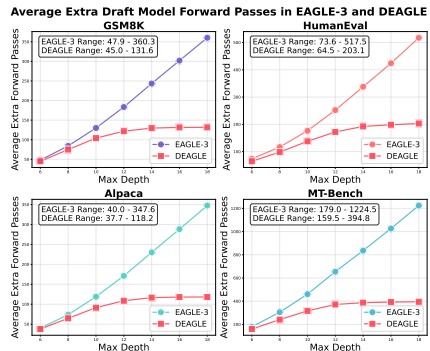

Figure 2: We use the comparison of draft model extra forward passes between EAGLE-3 and DEA-GLE to better quantify the effect of our proposed method. As shown in the graph, the proposed algorithm effectively controls the excessive forward passes as we increase the maximum depth.

Figure 2 presents the average number of draft model forward passes across different tree depths on four benchmarks. DEAGLE consistently gives fewer extra forward passes than EAGLE-3, especially as the maximum depth increases. On **GSM8K**, DEAGLE reduces the forward pass count from 360.3 (EAGLE-3) to 131.6 at depth 18, a **63% reduction**. Similarly, on **HumanEval**, the peak forward pass count drops from 517.5 to 203.1. The benefit is even more pronounced on **MT-Bench**, where EAGLE-3 performs up to 1224.5 forward passes per prompt at depth 18, while DEAGLE caps out at just 394.8—nearly a **3× reduction**. This shows that DEAGLE successfully identified and stopped exploration of low-confidence branches early, especially in conversational tasks with longer, more diverse responses. Across all tasks, DEAGLE maintains tight control of the tree growth without introducing too much overhead to sacrifice the overall speedup.

### 4.4 4.4 ABLATION STUDY ON STOPPING HEURISTICS

To understand the effectiveness of DEAGLE's voting-based stopping mechanism, we did an ablation study on its three heuristics: expectation, momentum ratio, and survival probability. We evaluate each of them independently as the sole stopping condition, and compare them to the full voting system. Table 4 reports the decoding speedup of each variant on MT-Bench using LLaMA3-8B across different tree depths at $T = 1$.

| Depth | MT-Bench | | | |
| | Exp | Momentum | Prob | Voting |
|---|---|---|---|---|
| 6 | 3.73x | 3.59x | 3.56x | **3.75x** |
| 8 | 3.85x | 3.77x | 3.76x | **3.95x** |
| 10 | 3.84x | 3.77x | 3.78x | **3.96x** |
| 12 | 3.90x | 3.69x | 3.78x | **3.92x** |
| 14 | 3.88x | 3.69x | 3.75x | **3.90x** |
| 16 | 3.87x | 3.63x | 3.75x | **3.89x** |
| 18 | 3.88x | 3.65x | 3.77x | **3.94x** |

Table 4: Speedup comparison of DEAGLE with different stopping strategies on LLaMA3-8B (MT-Bench, $T = 0$). The full voting strategy yields the highest efficiency across all tree depths.

### 5 CONCLUSION

In this work, we introduced DEAGLE, an enhanced speculative decoding framework that improves upon EAGLE-3 by adaptively controlling tree depth via a voting-based branch stopping mechanism. Unlike prior approaches that rely on fixed-depth trees, DEAGLE leverages three complementary factors: survival probability, momentum ratio, and expected acceptance length to decide when to stop tree expansion. Extensive experiments on Vicuna-13B, LLaMA3-8B, and LLaMA3-70B demonstrate that DEAGLE consistently outperforms EAGLE-3 in decoding speed across diverse benchmarks and decoding temperatures.

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
