# OpenReview forum: "DEAGLE: Token Tree with Dynamic Depth Will Further Benefit the Speculative Decoding"
_ICLR.cc/2026/Conference — Submitted to ICLR 2026_

### Official Review · Reviewer_B8kF · 2025-10-16

**Soundness:** 3
**Presentation:** 3
**Contribution:** 2
**Rating:** 4
**Confidence:** 5

**Summary:**

This paper proposes a token tree with dynamic depth when using SpD to accelerate LLMs. They show that the product of draft confidences along a token path can be a good heuristic for full-branch acceptance, and introduces a voting based early stopping mechanism that monitors the survival probability sum of the top-k leaves, and the expected accept length for the whole token tree. The experimental results show the effectiveness of the proposed method.

**Strengths:**

The paper is technically sound and easy to understand.

The theoretical proof show the usefulness of the method.

The experimental results show the effectiveness of the method.

**Weaknesses:**

The experimental results in Tab.1-3 is redundant. The author should report the best speedup compared to the baseline. Using different depth should be discussed in the ablation study.

The improvement is marginal when only compare the best speedup of DEAGLE and Eagle3.

The paper focuses on generating tree structure and improve the overall MAT to speedup the large language model. However, they do not report the framework they use. Here comes a problem that the method may not have such speedup on the popular inference framework such as vLLM. In fact, HuggingFace Transformers framework does not optimize the speed of LLMs very well, which makes the ratio of the latency of tree generation process smaller. When using vLLM framework where the operations in LLMs are optimized very well, the tree generation process will take more time and reduce the speedup.

The author should verify their method on such inference frameworks to show that their method is actually useful in reality.

**Questions:**

See weaknesses above.

---

> ### Author Response · Authors · 2025-11-20
> **Official Comment by Authors**
>
> Thank you so much for giving valuable thoughts.
>
> **Q1: The experimental results in Tab.1-3 are redundant**
>
> A: We put the experimental results with different maximum layers to show that our approach makes EAGLE-3 no longer depend on the maximum depth parameters. We want to show that the depth adaptive mechanism can choose the best depth based on the inputs.
>
> **Q2: The experimental results in Tab.1-3 are redundant**
>
> A: Please refer to the answer to Q2 for reviewer e5E2 and the answer to Q1 for reviewer TXEZ.
>
> **Q3: The experimental results in Tab.1-3 are redundant**
>
> A: We followed the setting of EAGLE-3 and didn't use any specific inference framework. The tree-based speculative decoding may not be compatible if we directly use those frameworks.

---

### Official Review · Reviewer_TXEZ · 2025-10-28

**Soundness:** 3
**Presentation:** 3
**Contribution:** 2
**Rating:** 2
**Confidence:** 5

**Summary:**

The paper presents a training-free enhancement to the EAGLE speculative decoding framework.  They show that survival probability
expectation is a good heuristic for the draft model to stop building the token tree, and propose DEAGLE, a dynamic depth control method. The experiments show the effectiveness of the proposed method.

**Strengths:**

1. This paper is technically sound and easy to understand.
2. Good theorical proof.

**Weaknesses:**

The experiments in Table 1 and Table 2 show no significant improvement over EAGLE3.

**Questions:**

See weakness above, speed improvement is important.

---

> ### Author Response · Authors · 2025-11-20
> **Official Comment by Authors**
>
> Thank you so much for providing valuable reviews.
>
> **Q1: No Significant Improvement**
>
> A: There are two reasons for the small improvement. First, the number you chose to compare is at the setting where the mamixum depth is set to 10, which is the optimal hyperparameter chosen by EAGLE-3. From the EAGLE-3 paper, the average acceptance length is 6.65. After ignoring the root token, the room for improvement is just 9 - 6.65 = 2.35. But as the depth gets deeper, the room for improvement will be greater, and you can see that the improvement is getting larger. Another reason is that the tree generation cost only takes around 10% of the overall computations when the depth is 12. In that case, even if we reduce 50% of tree generations, there will be only a 5% overall improvement. The value of our method will appear as the depth goes deeper. A more straightforward metric will be the average draft tree depth shown in Table 5 in Appendix D. The visualization in Figure 4 also shows the effectiveness of our approach. And, other than the proof, the key part of our approach is to free the constraints of the maximum depth. For example, for an unknown dataset, there is no need to find its optimal draft tree depth with our method, as it will adaptively determine the best tree depth for different inputs.

---

### Official Review · Reviewer_e5E2 · 2025-10-31

**Soundness:** 3
**Presentation:** 2
**Contribution:** 2
**Rating:** 4
**Confidence:** 4

**Summary:**

DEAGLE is a new speculative decoding framework designed as a training-free extension to EAGLE-3 to address the inefficiency of fixed-depth token trees. The paper formally proves that draft model confidence serves as an unbiased estimator for token acceptance, extending this theoretical basis from EAGLE-2 to the EAGLE-3 architecture. Based on this, DEAGLE introduces a novel voting-based early stopping mechanism that dynamically adjusts the tree expansion depth during inference. This mechanism monitors the top-k survival probability, the decay rate (momentum), and the estimated expected acceptance length of the token tree.

**Strengths:**

- The work provides the first formal proof that draft confidence acts as an unbiased estimator for token-level acceptance within the EAGLE-3 framework, grounding the dynamic decoding strategy in solid theory.

- DEAGLE utilizes a sophisticated three-factor voting mechanism (survival probability sum, momentum, and expected length) to effectively stop redundant branch expansion, significantly reducing unnecessary draft model forward passes.

**Weaknesses:**

1. Novelty Concerns: I have some concerns regarding the novelty of this paper, as it bears considerable similarity to the previous work, DDD[1]. Both works propose using Dynamic Tree Length for EAGLE, with the distinction that DDD is based on EAGLE2 while yours is based on EAGLE3. While DDD measures the total confidence of the entire beam by performing a log-sum-exp operation, your approach calculates the cumulative product (survival probability) for each leaf node, selecting the top-k most promising paths, and further incorporating Momentum and Expectation for voting. Nevertheless, the fundamental principle in both is the utilization of the draft model's probability for each draft token. Consequently, I consider your work to be somewhat incremental. However, you have enriched their methodology and provided corresponding theoretical foundations.

2. Marginal Performance Improvement: The performance gains are too marginal. For instance, in Table 1, at temperature=0, the SOTA improvement of DEAGLE over EAGLE3's SOTA is only 4.3% (4.37 to 4.56) on the MT-Bench and 3.5% (5.42 to 5.61) on the HumanEval.

3. Formatting Check: I recommend the authors carefully check the paper format. For example, there appears to be a duplicate section number at Line 410, which has "4.2" listed twice.

[1] Dynamic depth decoding: Faster speculative decoding for llms

**Questions:**

- As a suggestion for clarity, I recommend the authors add an "Average" column to Tables 1, 2, and 3. This column should report the average performance of EAGLE3 and DEAGLE across all four datasets. This would visually demonstrate the average performance of different methods across various depths.

- Some research[2] suggests that earlier tokens are more critical. I am curious whether there would be any effect if you were to increase the weight for the earlier tokens when calculating the survival probability.

- DDD shares the same goal as your work: proposing the use of a Dynamic length, but is based on EAGLE2, whereas yours is based on EAGLE3. I would like to know if you have conducted experiments to verify how their method performs when applied to EAGLE3. Specifically, would your proposed improvements be more effective than theirs when both are applied to the EAGLE3 framework?

- Why have the authors chosen not to report the Mean Accepted Length ($\tau$) in the experimental results, as was done in the EAGLE3 paper? I am keen to see the results for this metric as well.

I look forward to the authors' response and will adjust my score based on your clarification.

[2] Gumiho: A Hybrid Architecture to Prioritize Early Tokens in Speculative Decoding

---

> ### Author Response · Authors · 2025-11-20
> **Official Comment by Authors**
>
> Thank you so much for providing valuable insights. I will try to answer the questions you mentioned above.
>
> **Q1: Novelty Concerns**
>
> A: Yes, there are some similarities between this work and our work. However, just as mentioned in the related works part, the approach suggested by DDD was a naive method that directly uses the logprob sum from EAGLE's draft tree along with a threshold. That threshold-based method will limit its generalizability (e.g., for different inputs or datasets, the optimal threshold might change). What we are doing here is to reduce that dependency a little bit. And we also noticed that there is no proof of the observed relationships shown in EAGLE-2, so we started from the proof and gave a better heuristic for the adaptive draft tree generation. I would say half of the novelty will be from the proof, and now we know why EAGLE-2 is working mathematically.
>
> **Q2: Marginal Performance Improvement**
>
> A: The improvement is small because the maximum depth set by EAGLE-3 is the optimal hyperparameter, and it is really close to the average acceptance length. Therefore, the improvement is kind of limited. But, as we can see, our approach will determine the best depth, even if the hyperparameter is not optimal. Also, tree generation only takes around 10% of the computations during the generation when the depth is 12. Even if we reduced 50% of the unnecessary generations, the overall improvement might be just 5%. A better metric can be the average draft tree depth. In Figure 2 at the ablation study and Figure 3 in Appendix C, we can see that a large number of the layer generations are reduced after using our methods. And the average depth shown in Table 5 also shows a great improvement in reducing the tree depths. If there is a case where the tree generation takes the majority computations, the acceleration ratio number could be better.
>
> **Q3: Add an "Average" column to Tables 1, 2, and 3**
>
> A: Sure, I will add that part in the next version.
>
> **Q4: Increase the Weight for the Earlier Tokens**
>
> A: Yes, that is true. From our experiments, earlier tokens usually have higher probability scores. And if an early token gives a low score, its child token is likely to give a low score. Your idea of assigning an extra weight for tokens at different depths is interesting, and maybe I can try that. But things like how to design those weights and whether that will break the current proof (as we are using the expectation to estimate the tree depth) will take some time.
>
> **Q5: Conduct Experiments for DDD**
>
> A: Unfortunately, we can't find any code for DDD online, so we cannot replicate their results.
>
> **Q6: Why have the authors chosen not to report the Mean Accepted Length?**
>
> A: The Mean Accepted Length was used to evaluate the performance of the draft model. Since we are still using the EAGLE-3's framework and there is no improvement of the draft model itself, the Mean Accepted Length remains the same. The proposed method was intentionally set to overestimate the acceptance length a little bit, and there is no underestimated length case when we are collecting data for Figures 2,3,4. Therefore, we choose not to report those numbers.

---

> > ### Comment · Reviewer_e5E2 · 2025-11-23
> >
> > Thank you for your response, but my concerns remain largely unaddressed.
> >
> > My core concern is that the marginal performance improvement in the final speedup ratio (e.g., 3.5% to 4.3% at T=0 in Table 1) appears insufficient to justify the proposed complexity.
> >
> > The fundamental goal of speculative decoding is to accelerate the LLM inference. While you emphasize the significant reduction in redundant Draft Model forward passes, this improvement in computational efficiency ultimately fails to translate into a meaningful gain in the user-perceived speedup ratio.
> >
> > I understand that the tree generation cost may only account for 10% of the total computation. However, my question is: What is the practical utility of an optimization that maximally improves internal efficiency (e.g., 63% reduction in passes), yet only yields a trivial boost to the core metric of LLM acceleration (speedup ratio)?
> >
> > If maximizing computational efficiency still results in a marginal speedup, the proposed method's practical impact on addressing the LLM latency bottleneck is highly questionable.

---

### Meta-Review · Area_Chair_bRP8 · 2026-01-07

**Summary:**

The paper proposes a training-free speculative decoding framework that dynamically adjusts token tree depth based on EAGLE-3, leveraging draft model confidence and a voting-based early stopping mechanism, supported by theoretical analysis. Reviewers generally find the paper easy to understand (reviewers TXEZ and B8kF) and agree that it provides valid and sound theoretical proofs (all reviewers). However, all reviewers raise concerns about the marginal improvement, and reviewer e5E2 considers the novelty insufficient. In addition, some reviewers point out presentation issues, including formatting errors (reviewer e5E2) and problems with the experimental table (reviewer B8kF).

Most importantly, all three reviewers share a central concern: the paper fails to convincingly demonstrate or validate actual inference speed improvements, which are critical for speculative decoding methods. Reviewer e5E2 further notes that the authors’ response does not adequately address this issue. Since inference speedup is a core objective of speculative decoding, this weakness significantly undermines the paper’s impact and practical relevance. As a result, the reviewers collectively lean toward recommending rejection.

**Reviewer Concerns:**

I believe the novelty concern raised by reviewer e5E2 has been adequately addressed.

However, the shared concern among all three reviewers regarding the inference speedup of the proposed method remains insufficiently resolved. The authors should include quantitative analysis to clearly demonstrate and validate the LLM acceleration achieved by their framework.

**Reviewer Scores:**

- Reviewer e5E2: I believe the score would not change, as this reviewer considers their concerns to remain largely unaddressed.
- Reviewer TXEZ: I also do not expect a score change, since this reviewer is primarily concerned with inference speedup and finds no significant improvement over EAGLE-3. The authors do not provide sufficient additional evidence to substantiate their claimed improvements.
- Reviewer B8kF: Likewise, the score is unlikely to change, as the authors’ response appears insufficiently serious, evidenced by errors in the question summary.

---

### Decision · Program_Chairs · 2026-01-26

Reject